# MNCD: A New Tool for Classifying Parkinson’s Disease in Daily Clinical Practice

**DOI:** 10.3390/diagnostics12010055

**Published:** 2021-12-28

**Authors:** Diego Santos García, María Álvarez Sauco, Matilde Calopa, Fátima Carrillo, Francisco Escamilla Sevilla, Eric Freire, Rocío García Ramos, Jaime Kulisevsky, Juan Carlos Gómez Esteban, Inés Legarda, María Rosario Isabel Luquín, Juan Carlos Martínez Castrillo, Pablo Martínez-Martin, Irene Martínez-Torres, Pablo Mir, Ángel Sesar Ignacio

**Affiliations:** 1Unidad de Trastornos de Movimiento, Servicio de Neurología, CHUAC, Complejo Hospitalario Universitario de A Coruña, 15009 A Coruña, Spain; 2Neurología, Hospital San Rafael, 15009 A Coruña, Spain; 3Departamento de Neurología, Hospital General Universitario de Elche, 03203 Elche, Spain; mariaalsa@hotmail.com (M.Á.S.); dr.freyre@gmail.com (E.F.); 4Unidad de Trastornos del Movimiento, Servicio de Neurología, Hospital Universitari de Bellvitge, 08907 Barcelona, Spain; mcalopa@bellvitgehospital.cat; 5Unidad de Trastornos del Movimiento, Servicio de Neurología y Neurofisiología Clínica, Instituto de Biomedicina de Sevilla, Hospital Universitario Virgen del Rocío, 41013 Sevilla, Spain; fmcarrillog@gmail.com (F.C.); pmir@us.es (P.M.); 6Centro de Investigación Biomédica en Red sobre Enfermedades Neurodegenerativas CIBERNED, 28031 Madrid, Spain; jkulisevsky@santpau.cat (J.K.); pmm650@hotmail.com (P.M.-M.); 7Unidad de Trastornos del Movimiento, Servicio de Neurología, Hospital Universitario Virgen de las Nieves, Instituto de Investigación Biosanitaria (ibs.Granada), 18013 Granada, Spain; fescamilla@hotmail.com; 8Departamento de Neurología, Hospital IMED Elche, 03203 Elche, Spain; 9Instituto de Investigación Sanitaria San Carlos (IdISCC), Hospital Clínico San Carlos, 28040 Madrid, Spain; garciaramosg@yahoo.es; 10Departamento de Neurología, Unidad de Trastornos del Movimiento, Hospital de la Santa Creu i Sant Pau, 08041 Barcelona, Spain; 11Servicio de Neurología, Hospital Cruces, 48903 Bilbao, Spain; juancarlos.gomezesteban@gmail.com; 12Hospital Universitario Son Espases, 07120 Palma, Spain; ines.legarda@ssib.es; 13Departamento de Neurología, Clínica Universidad de Navarra, Instituto de Investigación Sanitaria de Navarra, 31008 Pamplona, Spain; rluquin@unav.es; 14Servicio de Neurología, Hospital Ramón y Cajal, 28034 Madrid, Spain; jcmcastrillo@gmail.com; 15Unidad de Trastornos del Movimiento, Servicio de Neurología, Hospital Universitario y Politècnico La Fe, 46026 Valencia, Spain; irenemto@hotmail.com; 16Unidad de Trastornos del Movimiento, Servicio de Neurología, CHUS (Complejo Hospitalario Universitario de Santiago de Compostela), 15706 A Coruña, Spain; angel.sesar.ignacio@sergas.es

**Keywords:** cognition, dependency, non-motor symptoms, motor symptoms, Parkinson’s disease

## Abstract

**Background and objective**: Parkinson’s disease (PD) is a clinically heterogeneous disorder in which the symptoms and prognosis can be very different among patients. We propose a new simple classification to identify key symptoms and staging in PD. **Patients and Methods**: Sixteen movement disorders specialists from Spain participated in this project. The classification was consensually approved after a discussion and review process from June to October 2021. The TNM classification and the National Institutes of Health Stroke Scale (NIHSS) were considered as models in the design. **Results**: The classification was named MNCD and included 4 major axes: (1) motor symptoms; (2) non-motor symptoms; (3) cognition; (4) dependency for activities of daily living (ADL). Motor axis included 4 sub-axes: (1) motor fluctuations; (2) dyskinesia; (3) axial symptoms; (4) tremor. Four other sub-axes were included in the non-motor axis: (1) neuropsychiatric symptoms; (2) autonomic dysfunction; (3) sleep disturbances and fatigue; (4) pain and sensory disorders. According to the MNCD, 5 stages were considered, from stage 1 (no disabling motor or non-motor symptoms with normal cognition and independency for ADL) to 5 (dementia and dependency for basic ADL). **Conclusions**: A new simple classification of PD is proposed. The MNCD classification includes 4 major axes and 5 stages to identify key symptoms and monitor the evolution of the disease in patients with PD. It is necessary to apply this proof of concept in a properly designed study.

## 1. Introduction

Parkinson’s disease (PD) is a complex disorder with a wide variety of symptoms that have a negative impact on patients’ quality of life (QoL) and independence for activities of daily living (ADL) [1]. Importantly, PD is clinically heterogeneous, since symptoms and prognosis can be very different among patients [2], so that having a simple classification, which could properly inform about key symptoms at different stages of the disease, would be of great importance. Based on the classic motor symptoms of the disease, the Hoehn and Yahr (H&Y) scale is used to describe the progression of PD [3]. The scale was originally described in 1967 and included stages 1 through 5. It has been modified with the addition of stages 1.5 and 2.5 to help describe the intermediate course of the disease [4]. However, although H&Y is frequently used in clinical practice, in part because it is very simple and easy to interpret, the information it provides is limited to the motor stage. The importance of non-motor symptoms (NMS) has increased over the last years because they are frequent and disabling, and impact negatively on patients’ QoL independently of other variables [5]. Recently, a new classification combining the H&Y and the non-motor symptoms scale (NMSS) has been suggested [6]. According to this scale (H&Y-NMSS), patients with a greater global NMS burden but a lower H&Y stage can have a worse QoL than patients with a higher H&Y stage but a lower NMS burden [7]. Therefore, NMS are frequent even in early stages of PD and impact significantly on QoL.

Ideally, a classification for a neurodegenerative disease should include key symptoms and biomarkers that, if present, imply an important therapeutic decision or have a great prognostic value [8]. Some key symptoms in PD are motor fluctuations, dyskinesia, dysphagia, freezing of gait (FOG), falls, cognitive impairment, psychosis, impulse control disorder, major depression, or autonomic symptoms [9]. In particular, many of these symptoms are present in what has been called advanced PD [10]. However, the definition of advanced PD is too broad and includes patients starting with disabling motor fluctuations at one extreme and patients with dementia in an advanced palliative stage at the other. The CDEPA questionnaire is a valid, reliable, and useful instrument for screening advanced PD [11]. In practice, however, it is somewhat complex and takes time to use, given the controversial, high variability in the definition of advanced PD. Moreover, labeling a patient with advanced PD could have a negative connotation in certain circumstances, such as the COVID pandemic [12]. Alternatively, 5-2-1 criteria (≥5 times oral levodopa tablet taken/day; ≥2 h of OFF time/day; ≥1 h/day of troublesome dyskinesia) is a simple and useful tool for early identification of advancing disease [13,14] but its applicability in all PD patients is uncertain (e.g., older PD patients without clear fluctuations but axial or cognitive symptoms). More recently, MANAGE-PD has been proposed as a simple, clinician-reported, screening web-based tool (https://managepd.com/, accessed on 23 December 2021) to identify patients with PD inadequately controlled on oral medications [15]. Again, the application of this tool takes time and does not provide a straightforward clinical picture of a patient’s condition, which limits its value in routine clinical practice.

In this context, we propose a new simple classification that makes it possible to identify key symptoms and monitor the evolution of the disease in patients with PD, including the definition of different stages.

## 2. Material and Methods

Sixteen movement disorders specialists from Spain participated in this project. A first classification proposal was developed by one of them (DSG) in May 2021. It was initially reviewed by the rest of the neurologists. On 21 June 2021, a telematic meeting of about 2 h was held in which all the points of the project and the classification were discussed by the neurologists. A second version of the classification was proposed and finally, it was consensually approved in October 2021.

The main points approved by all the participants are shown in Table 1. Importantly, it was considered that the purpose of this new classification is to provide a quick visual interpretation of a patient’s condition, both for the doctor who follows him/her regularly and for any other specialist. Moreover, the classification may be used at each clinical visit to monitor the evolution of the patient in terms of symptoms control and disease progression, with a staging established on this basis. Specifically, the TNM classification [16] and the National Institutes of Health Stroke Scale (NIHSS) [17] used in oncology and in vascular neurology, respectively, were considered in the design as models.

The project was planned in two phases. The first phase consisted of reporting a new classification and staging of PD agreed upon by expert neurologists in movement disorders through scientific publication in 2021. The second part would contemplate the development of a study to apply the new classification in patients in clinical practice from the year 2022. A timeline of the project development deadlines is shown in Figure 1.

## 3. Results

In the proposed classification, 4 major axes were considered: (1) **M**otor symptoms; (2) Non-motor symptoms; (3) **C**ognition; (4) **D**ependency for ADL. Using the TNM classification as a model, the classification was named as **MNCD**, such that each letter is the first letter of each major axis.

### 3.1. First Axis

The first axis considered was **M**otor symptoms (**M**), subdivided into 4 defined sub axes: (1) motor fluctuations; (2) dyskinesia; (3) axial symptoms; (4) tremor.

For each of the axes, the presence or absence of the symptom is established. Each sub-axis is scored as 1 when symptoms are present and are clinically relevant at the discretion of the neurologist and 0 if there are no symptoms or they are quite minor and not relevant. In the MNCD classification, the number, from 0 to 4, is placed as a subscript next to M: M_0_ (no sub-axis with symptoms); M_1_ (1 sub-axis with symptoms); M_2_ (2 sub-axes with symptoms); M_3_ (3 sub-axes with symptoms); M_4_ (all sub-axes with symptoms). All types of motor fluctuations (wearing-off, early-morning akinesia, sudden-off, on-off, no-on, delayed-on) and dyskinesia (peak-dose, benefit of dose, diphasic, off period dystonia) should be considered [18]. Axial symptoms for taking in care include FOG, postural instability, trunk posture alterations, dysphagia and dysarthrophonia [19]. Any type of significant disabling tremor (resting, postural, etc.) should be scored as 1.

Using the NIHSS as a model, it is indicated which axis is showing symptoms: e.g., M_2_ (1100), a patient with motor fluctuations and dyskinesia but neither significant axial symptoms nor significant tremor; M_1_ (0010), a patient with at least one significant axial symptom (e.g., FOG) but without motor fluctuations, dyskinesia and significant tremor; M_1_ (0001), a patient with only tremor as a significant disabling motor symptom. In summary, the information about axis 1 would be displayed as M_0-4_ (_ _ _ _) (Figure 2).

### 3.2. Second Axis

The second axis considered was **N**on-motor symptoms (**N**), subdivided into 4 defined sub axes: (1) neuropsychiatric symptoms; (2) autonomic dysfunction; (3) sleep disturbances and fatigue; (4) pain and sensory disorders.

Again, for each of the axes the presence or absence of relevant symptoms is established. In the MNCD classification, the number, from 0 to 4, is placed as a subscript next to N: N_0_ (no sub-axis with symptoms); N_1_ (1 sub-axis with symptoms); N_2_ (2 sub-axes with symptoms); N_3_ (3 sub-axes with symptoms); N_4_ (all sub-axes with symptoms). Each sub-axis is scored as 1 when symptoms are clinically relevant at the discretion of the neurologist; in sub-axis 1 (neuropsychiatric symptoms), depression, anxiety, apathy, visual hallucinations, psychosis, impulse control disorder, etc. [20]; in sub-axis 2 (autonomic dysfunction), cardiovascular (orthostatic hypotension, syncope, supine hypertension, postprandial hypotension), gastrointestinal (constipation, obstipation, nausea, vomiting, early satiety), urinary (urgency, frequency, noctoria, urinary retention, incontinence), sexual (sexual dysfunction, impotence) and other dysautonomic symptoms (anhidrosis, compensatory hyperhidrosis, venous pooling, acral color changes, dry mouth, etc.) [21]; in sub-axis 3 (sleep disturbances and fatigue), physical fatigue, insomnia, fragmented sleep, excessive daytime sleepiness (EDS), periodic limb movements in sleep (PLMS), restless legs syndrome (RLS), sleep apnea, rapid eye movement sleep behavior disorder (RBD), etc. [22]; in sub-axis 4 (pain and sensory disorders), pain and related symptoms, abnormal sensations, vestibular deficits, hyposmia, color vision deficits, etc. [23].

Again, using the NIHSS as a model, MNCD classification would indicate which axis is showing symptoms: e.g., N_2_ (1100), a patient with at least one neuropsychiatric symptom (e.g., depression) and one dysautonomic symptom (e.g., orthostatic hypotension) but without relevant symptoms regarding fatigue, sleep, pain and sensory symptoms; N_2_ (1001), a patient with anxiety and pain but no other significant NMS; N_3_ (1110), a patient with visual hallucinations, constipation, and EDS such as disabling NMS. In summary, the information about axis 2 would be displayed as N_0-4_ (_ _ _ _) (Figure 2).

### 3.3. Third Axis

The third axis considered was **C**ognition (**C**). Cognition was considered by itself as a single axis (without sub-axes) with 3 excluding options (only one option is possible): 0, normal cognition; 1, mild cognitive impairment; 2, dementia. In clinical and research settings, the term ‘mild cognitive impairment’ is applied to PD patients who present cognitive complaints and whose neuropsychological examinations confirm the deficits, but PD dementia criteria cannot be fulfilled due to the lack of overt functional decline related to cognitive impairment [24]. In broad terms, mild cognitive impairment can be defined as a cognitive decline from a previous performance baseline, that is considered abnormal for the patient’s age, but with retention of normal daily functioning [25]. In the MNCD classification, the number, from 0 to 2, is placed as a subscript next to C: C_0_ (normal cognition); C_1_ (mild cognitive impairment); C_2_ (dementia). The information about axis 3 would be presented as C_0-2_ (_) (Figure 2).

### 3.4. Fourth Axis

The fourth axis considered was **D**ependency for ADL (**D**). Two major groups of ADL have been reported: ‘basic’ activities related to self-care, such as bathing, dressing, eating, voluntary control of sphincters, grooming and walking; and ‘instrumental’ activities, such as light housework, preparing meals, taking medications, shopping for groceries or clothes, using the telephone and managing money [26,27]. Dependency was considered by itself as a single axis (without sub-axes) with 3 excluding options (only one option is possible): 0, independence for ADL; 1, dependency for instrumental but not for basic ADL; 2, dependency for basic ADL. In the MNCD classification, the number, from 0 to 2, is placed as a subscript next to D: D_0_ (independency for ADL); D_1_ (dependency for instrumental ADL); D_2_ (dependency for basic ADL). The information about axis 4 would be presented as D_0–2_ (_) (Figure 2).

### 3.5. How to Show the MNCD Classification and Interpret It

The classification would be presented as: M_0–4_N_0–4_C_0–2_D_0–2_ (_ _ _ _/_ _ _ _/_/_)–X. Firstly, the 4 letters are shown with the corresponding number as a subscript. Secondly, the sequence in order, with the number corresponding to each sub-axis inside parentheses, with a separation between major axes. Finally, a dash followed by a number that would indicate the time from PD symptoms onset. Some examples are shown in Table 2.

### 3.6. Stages of Disease According to the MNCD Classification

According to the MNCD and considering the TNM classification used in oncology as a model, 5 stages are considered (Table 3 and Figure 2). In stage 1, the patient has no relevant motor and non-motor symptoms, being independent for basic ADL and without cognitive impairment. In stage 2, there is at least 1 motor symptom or 1 NMS scoring in the MNCD classification, but there is neither cognitive impairment nor dependency for ADL. In stage 3, there is mild cognitive impairment (C = 1) and/or dependency for instrumental ADL (D = 1); the score on axes 1 and 2 could be ≥1. In stage 4, the patient is dependent for basic ADL (D = 2). Finally, in stage 5, the patient has dementia (C = 2) and is functionally dependent for basic ADL (D = 2). Exemplifying this classification in different stages, the stages of the previous examples are shown in Table 2. The MNCD classification could be useful for monitoring the progression of PD from the very beginning of the disease (Figure 3).

## 4. Discussion

The present manuscript proposes a new classification of PD that is intended to be both simple and useful in clinical practice. It has been elaborated and agreed upon by a group of sixteen movement disorders specialists from Spain with proven experience in the management of PD and includes 4 major axes: (1) motor symptoms; (2) NMS; (3) cognition; (4) independence loss for ADL. Furthermore, it is intended to be a useful instrument to monitor the progression of the disease and to be able to categorize the patient in a different evolutionary stage. To our knowledge, this is the first PD classification that takes into account key aspects of the disease such as axial symptoms, NMS, cognitive problems and their impact on disability and QoL.

The group of experts reached a consensus to consider the 4 axes described as the capital axes of the classification. This is because all axes and sub-axes refer to key symptoms due to their impact on the patient and/or caregiver, their prognostic value and/or their importance when deciding on a specific therapeutic attitude. Motor fluctuations appear early in PD patients and frequently are underdiagnosed [28]. Importantly, the presence of motor fluctuations in early PD patients is associated with greater NMS burden and dependency for ADL and a worse QoL [29], representing a major criteria for eligibility to advanced treatments [30]. Moreover, observational studies have shown that more than 50% of PD patients treated with levodopa for more than 5 years develop levodopa-induced dyskinesia [31]. From a clinical point of view, the presence of dyskinesia is a turning point since its presence limits further increase of medication dosage and the evaluation of new strategies [32]. Axial symptoms associated with PD, such as FOG, postural instability, trunk posture alterations, dysphagia, and dysarthrophonia, have a significant impact on patients’ QoL and increase the risk of complications (i.e., falls, aspiration, etc.), hospitalization, institutionalization, functional dependency and mortality [33,34,35]. Moreover, these symptoms are poorly responsive to dopaminergic drugs and surgical therapies. Finally, although tremor dominant motor phenotype has a slower rate of progression with less deterioration regarding QoL [36], refractory severe tremor can be very disabling and, in the absence of contraindication, can improve with deep brain stimulation (DBS) [37] or high intensity focused ultrasound (HIFU) [38]. In summary, it is crucial to identify the presence of all these motor symptoms in PD patients throughout the course of the disease.

On the other hand, with the growing awareness of NMS in PD has come the realization that these non-motor features play a tremendously important, and sometimes dominant, role in the management and even the diagnosis of the disorder [22]. It is well known that NMS impact negatively on patient’s QoL and in some cases, such as dementia or psychosis, there is an increased risk of institutionalization and greater patient dependence for ADL and caregiver burden [5,7,39,40]. Prospective studies have demonstrated that the progression of NMS impacts on patient’s QoL. An increase in ≥5 and ≥10 points of BDI-II (Beck-Depression Inventory—II) and NMSS (Non-Motor Symptoms Scale) total score multiplied the probability of presenting clinically significant health related QoL impairment by 5 and 8, respectively, in 500 PD patients from Spain after a 2 year-follow up [41]. Therefore, it is crucial to identify the presence of NMS, quantify their severity, and analyze their short- and long-term progression. In the MNCD classification, NMS were considered one major axis (N), but cognition © was considered as a distinct axis in itself, given the implications it has for treatment and prognosis. Cognitive impairment must be detected early, and it should even be taken into account that subjective cognitive decline seems to predict future deterioration in cognitively normal PD patients [42]. The classification of the NMS into 4 sub-axes was carried out based on the literature and the usual way of grouping these symptoms [1,2,19,20,21,22]. The classification allows for the identification, if there are any relevant NMS, and of its particular type which is important from a clinical management point of view [43]. Ideally, it should be useful for the monitoring outcome or even the response to a therapeutic intervention (i.e., depression, pain, etc.). Finally, dependency for ADL was considered as the fourth axis. In a neurodegenerative disease such as PD, one of the objectives is to delay the progression of the disability for as long as possible and maintain patient independence for ADL. Loss of such independence represents another turning point including the repercussions at the personal and social level in the patient’s life. Loss of functional independence leads to caregiver burden, high resource use, institutionalization, comorbid complications, worse QoL and increased risk of death [44], and is also considered an important outcome of progression [45]. Very recently, it was reported that autonomy for ADL worsened in 507 PD patients compared to 124 controls from a Spanish cohort and that cognitive impairment, gait problems, fatigue, depressive symptoms, more advanced disease, and a non-tremor phenotype were independent predictors of functional dependency after a 2-year follow-up [34]. Significantly, all these symptoms are included in the MNCD classification in a very clear, simple and direct way to see and interpret.

As was mentioned in the introduction, other classifications for PD have been reported [3,4,6,9,11,13,14,15] and some are frequently used in clinical practice [3,4]. Motor symptoms are usually assessed by visual examination of motor tasks and semi-quantitative rating scales such as the H&Y [3,4], the Unified Parkinson’s Disease Rating Scale (UPDRS) [46], or more recently, the Movement Disorder Society—Unified Parkinson’s Disease Rating Scale (MDS-UPDRS) [47]. Motor assessment tools will probably be used in the future for PD diagnostic and progression [48]. However, the new devices are probably unlikely to be implemented in clinical practice in the coming years and, in general, to collect motor information [49]. With regard to NMS, recent clinical and neurobiological research suggests the existence of discrete non motor subtypes in PD: Park Cognitive; Park Apathy; Park Depression/Anxiety; Park Sleep; Park Pain; Park Fatigue; Park Autonomic [50]. The well-known motor phenotype classification is the one usually employed. However, combining both motor and NMS could be of great interest and usefulness [6,7]. Very recently, Brendel et al. [51], using data from the BioFIND cohort with clustering analysis, identified three unique subtypes: subtype I, characterized by mild symptoms, both motor and non-motor; subtype II, characterized by an intermediate severity, with a high tremor score and mild non-motor symptoms; and, subtype III, with more severe motor and non-motor symptoms. Although different biological markers may be useful in the future when classifying PD for its management (i.e., genetic, neuroimaging, biochemical, etc.) [52], in the Brendel study there was no clear difference in demographics, biomarker levels, and genetic risk scores. In a very simple and visual classification, our proposal includes not only motor and non-motor symptoms but also cognition as a key aspect, due to its implications for disability for daily activities. Unlike other proposals that are based on categorizing whether patients are in an advanced stage of the disease or not [11,13,14,15,16], the MNCD classification can be applied from the beginning and allows for the monitoring of evolutionary changes throughout the course of the disease, with clearly defined stages. Moreover, this classification is applicable in each individual case in a simple way and is not the result of a cluster analysis used in many patients, which is sometimes difficult to transfer to the individual case. In other words, the MNCD classification is a simple classification that alerts to the presence of the most important symptoms and complications in order to propose interventions, both at a transversal and longitudinal level. Furthermore, as it is a simple classification, it could be applied by the general neurologist, which is important because many patients with PD are treated by the general neurologist in Spain and other countries [53,54]. However, this is only a proposal and it is necessary to test its utility in clinical practice. Because scoring in the sub axes will be conditioned by the evaluator’s perception of the importance of the symptoms, it will also be necessary to know what the inter- and intra-observer variability is when using the classification. As explained, the objective of phase 2 of this project will be to prove its applicability in a daily clinical practice setting.

In conclusion, a new, simple classification of PD is proposed in this manuscript by sixteen experts on PD from Spain. The MNCD classification includes 4 major axes (motor; non-motor; cognition; dependency) and 5 stages to identify key symptoms and monitor PD progression. It is necessary to apply this proof of concept to the design of a study in order to verify the usefulness of the MNCD classification in daily clinical practice.

## Figures and Tables

**Figure 1 diagnostics-12-00055-f001:**
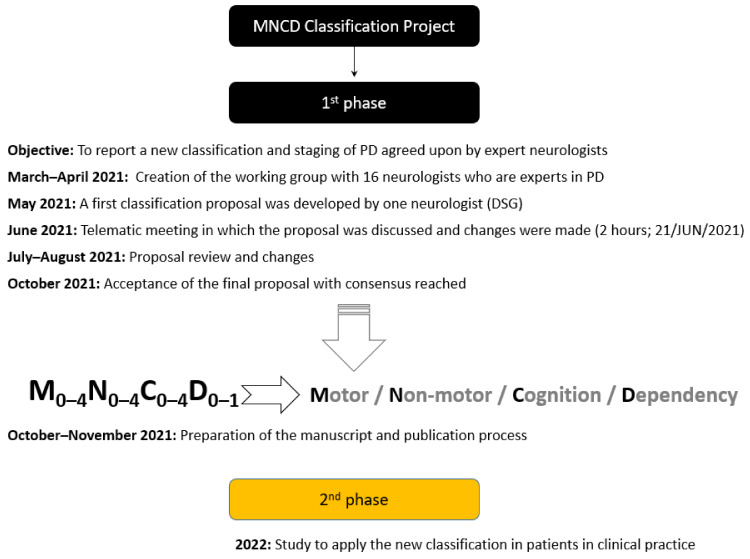
Flow chart on the phases and objectives of the project. PD, Parkinson’s disease.

**Figure 2 diagnostics-12-00055-f002:**
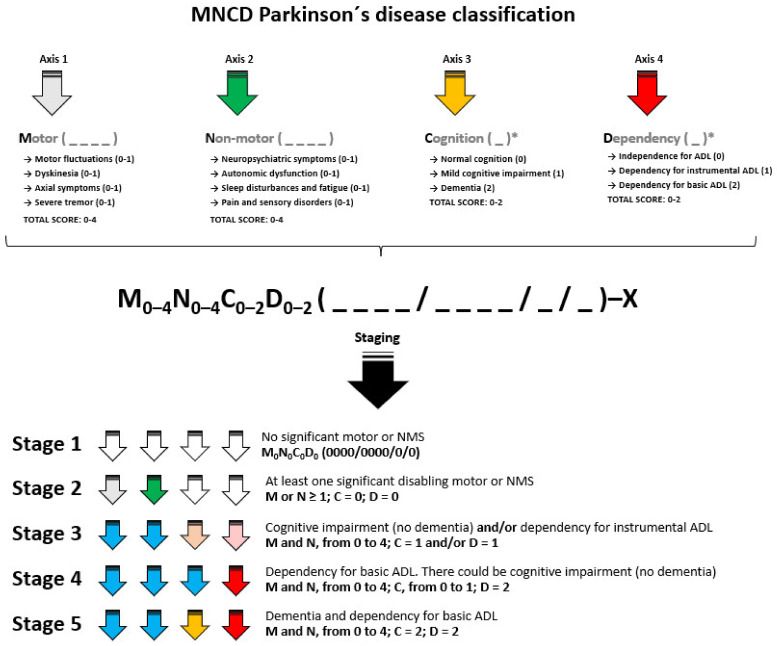
MNCD PD classification, showing the 4 major axis with their sub-axes and stages from 1 to 5. In Staging, an arrow with the same color as in the upper part indicates that there is a relevant symptomatology regarding the axis. For axes 3 and 4, light color indicates milder symptoms (mild cognitive impairment for axis 3 and dependency for instrumental ADL for axis 4) and dark color for a more severe affectation (dementia for axis 3 and dependency for ADL for axis 4). When the arrow is blue, it indicates that there may or may not be relevant symptoms related to the axis. *, it is considered by itself a single axis (without sub-axes) with excluding options. ADL, activities of daily living.

**Figure 3 diagnostics-12-00055-f003:**
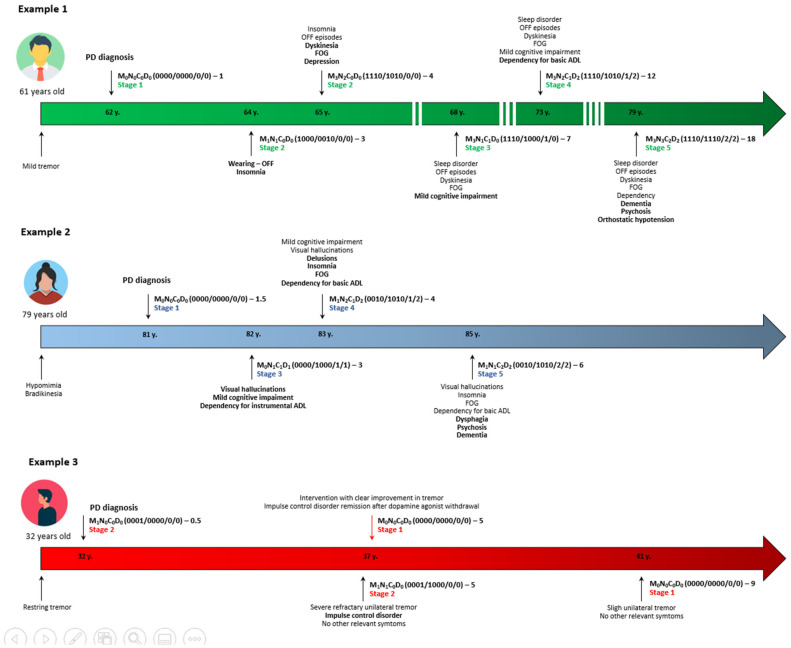
Three examples about the use of the MNCD classification are shown in PD patients with very different characteristics and outcomes (age from symptoms onset, phenotype, etc.). Example 1 represents a patient who developed dementia in the long-term; example 2, an elderly woman at symptoms onset who developed dementia and functional dependency in the short-term; example 3, a very young patient with an asymmetric tremor dominant benign form who improved after intervention and remained very stable after several years of evolution. FOG, freezing of gait; PD, Parkinson’s disease.

**Table 1 diagnostics-12-00055-t001:** The main points approved by the participating expert neurologists regarding the new PD classification are shown.

(1) It is considered necessary and of interest to propose a new simple and useful classification by axes of PD. (2) The number of axes should not be excessive (simplification is a prerequisite for usefulness). (3) Each axis must be based on a key aspect of the disease. (4) Each axis can then have different sub-axes. (5) The way to evaluate each axis must be simple so that the classification as a whole can be simple as well. (6) Cognitive problems constitute a key symptom and should be an axis on its own. (7) Loss of independence of autonomy for activities of daily living is a key symptom and should be an axis on its own. (8) Based on the information collected, a PD classification and PD staging could be defined. (9) The data for the classification will be obtained from the direct evaluation of the patient in clinical practice, either through direct interview and assessment by the evaluating neurologist or through the use of clinical scales. (10) The purpose of this classification is to enable a quick visual interpretation of the patient’s condition, both for the neurologist who follows them regularly and for other specialists. (11) The TNM classification used in oncology and the National Institutes of Health Stroke Scale (NIHSS) in vascular neurology were considered as a model in the design. (12) Based on the classification, a staging will be established. (13) The classification may be used at each clinical visit to monitor the evolution of the patient in terms of symptoms control and disease progression.

PD, Parkinson’s disease.

**Table 2 diagnostics-12-00055-t002:** Examples about the MNCD application.

**Example 1** • A 49 year old man, 6 months from symptoms onset with stage 1 of H&Y and Unified Parkinson’s Disease Rating Scale—part III score of 7 points, without cognitive impairment and any significant disabling motor or NMS and independent for ADL. → M_0_N_0_C_0_D_0_ (0000/0000/0/0)–5. → Stage 1. **Example 2** • A 50 year old woman, 2 years from symptoms onset with significant disabling resting tremor on the right part of the body but without axial symptoms, motor fluctuations, dyskinesia or any relevant NMS, with normal cognition and independent for ADL. → M_1_N_0_C_0_D_0_ (0001/0000/0/0)–2. → Stage 2. **Example 3** • A 70 year old woman, 6 years from symptoms onset with motor fluctuations, dyskinesia, FOG, depression, visual hallucinations, mild cognitive impairment and dependency for instrumental ADL. → M_3_N_1_C_1_D_1_ (1110/1000/1/1)–6. → Stage 3. **Example 4** • A 79 year old man, 4 years from symptoms onset with PIGD (postural instability gait difficulty) motor phenotype with FOG and falls but without motor complications, and severe RBD, pain and depression, without cognitive impairment but with loss of independence for basic ADL. → M_1_N_3_C_0_D_2_ (0010/1011/0/2)–4. → Stage 4. **Example 5** • A 69 year old woman, 20 years from symptoms onset with motor fluctuations without dyskinesia, FOG, falls, dysphagia, visual hallucinations with psychosis, orthostatic hypotension, severe insomnia, dementia, and dependency for ADL. → M_2_N_3_C_2_D_2_ (1010/1110/2/2)–20. → Stage 5.

**Table 3 diagnostics-12-00055-t003:** Stages in PD according to the MNCD classification.

**Stage 1** • The patient meets criteria of Parkinson’s disease but neither significant disabling motor or non-motor symptoms and there is no cognitive impairment. The patient is functionally independent for activities of daily living. • **MNCD classification is M_0_N_0_C_0_D_0_ (0000/0000/0/0).** **Stage 2** • The patient has at least one significant disabling motor or non-motor symptom. There is no cognitive impairment. The patient is functionally independent for instrumental and basic activities of daily living. • **MNCD classification: M or N ≥ 1; C = 0; D = 0.** **Stage 3** • There is cognitive impairment (no dementia) and/or dependency for instrumental ADL. • **MNCD classification: M and N, from 0 to 4; At least, C or D = 1.** **Stage 4** • The patient is functionally dependent for basic activities of daily living. There could be mild cognitive impairment but not dementia. • **MNCD classification: M and N, from 0 to 4; C, from 0 to 1; D = 2.** **Stage 5** • The patient is functionally dependent for basic ADL and there is dementia. • **MNCD classification: M and N, from 0 to 4; C = 2; D = 2.**

## Data Availability

Not appicable.

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
