# Peer review of "MNCD: A New Tool for Classifying Parkinson’s Disease in Daily Clinical Practice"

_diagnostics, 2021, doi:10.3390/diagnostics12010055_

Round 1

Reviewer 1 Report

The diagnistic approach proposed is very usefull and urgently needed. The new classification promises to be very helpful, as it systematically includes non-motor symptoms, cognitive problems, and quality of life (dependency)

There are a few question. The proposed scale might produce numerous false-positive results, since the weight of non-motor symptoms are high scored altogether. So, the cautions should be formulated in advance, to go together with te scale

Author Response

The diagnostic approach proposed is very usefull and urgently needed. The new classification promises to be very helpful, as it systematically includes non-motor symptoms, cognitive problems, and quality of life (dependency).

There are a few question. The proposed scale might produce numerous false-positive results, since the weight of non-motor symptoms are high scored altogether. So, the cautions should be formulated in advance, to go together with the scale.

AUTHORS – Thank you very much for your comment. We agree with you. In Discussion, a comment about this has been added: “However, this is only a proposal and it is necessary to test its utility in clinical practice. Because scoring in the sub axes will be conditioned by the evaluator's perception of the importance of the symptoms, it will also be necessary to know what the inter- and intra-observer variability is when using the classification”.

Reviewer 2 Report

The paper is of interest, and well written. It well describes PD motor symptoms and contains the relevant sections, references, and 2 tables ad 3 figures it presents MNCD: A New Tool for Classifying Parkinson´s Disease in Daily Clinical Practice.

The authors are affiliated with a Spanish university. 

The new disease measure scale is composed of motor, non motor cognition and dependency scales. 

it also describes 

MANAGE-PD has been proposed as a simple, clinician-reported, screening web-based tool (https://managepd.com/) to identify patients with PD inadequately controlled on oral medications 

 Sixteen movement disorders specialists from Spain participated in this project. 

You can add citation to the following paper

Deep brain stimulation induces rapidly reversible transcript changes in Parkinson's leucocytes. Soreq L, Bergman H, Goll Y, Greenberg DS, Israel Z, Soreq H.J Cell Mol Med. 2012 Jul;16(7):1496-507. doi: 10.1111/j.1582-4934.2011.01444.x. (which described the UPDRS disease rating scale).   Thank you! good luck 

Author Response

The paper is of interest, and well written. It well describes PD motor symptoms and contains the relevant sections, references, and 2 tables ad 3 figures it presents MNCD: A New Tool for Classifying Parkinson´s Disease in Daily Clinical Practice.

The authors are affiliated with a Spanish university.

The new disease measure scale is composed of motor, non motor cognition and dependency scales.

it also describes

MANAGE-PD has been proposed as a simple, clinician-reported, screening web-based tool (https://managepd.com/) to identify patients with PD inadequately controlled on oral medications

 Sixteen movement disorders specialists from Spain participated in this project.

You can add citation to the following paper

Deep brain stimulation induces rapidly reversible transcript changes in Parkinson's leucocytes. Soreq L, Bergman H, Goll Y, Greenberg DS, Israel Z, Soreq H.J Cell Mol Med. 2012 Jul;16(7):1496-507. doi: 10.1111/j.1582-4934.2011.01444.x. (which described the UPDRS disease rating scale).   Thank you! good luck

AUTHORS – Thank you very much for your comment. Sorry, this paper is about a very specific topic not related with the present manuscript. Are you sure this is the reference? We would really like to add it but we don´t find the context in which to do it.
